# The Arteries of the Encephalon Base in Caracal (*Caracal caracal*; Felidae; Carnivora)

**DOI:** 10.3390/ani13243780

**Published:** 2023-12-07

**Authors:** Maciej Zdun, Aleksander F. Butkiewicz, Marcin Zawadzki

**Affiliations:** 1Department of Basic and Preclinical Sciences, Nicolaus Copernicus University in Torun, Lwowska 1, 87-100 Torun, Poland; 304242@stud.umk.pl (A.F.B.); mzawadzki@umk.pl (M.Z.); 2Department of Animal Anatomy, Poznan University of Life Sciences, Wojska Polskiego 71C, 60-625 Poznań, Poland

**Keywords:** angiology, animal anatomy, encephalon vascularization, Felidae, wild cats

## Abstract

**Simple Summary:**

The caracal (*Caracal caracal*) is a medium-sized wild cat native to Africa, the Middle East, Central Asia, and arid areas of Pakistan and northwestern India. This study discusses the vascularization of the encephalon base in this species based on our research and its comparison with relevant scientific articles.

**Abstract:**

This study represents the comprehensive anatomical analysis of the arterial circulation at the base of the encephalon in caracal (*Caracal caracal*), a member of the Felidae family. Caracals are found in various environments in Africa and Asia, and their conservation status is threatened by hunting and habitat loss. This study was conducted on 14 post-mortem specimens obtained from zoos. Three different methods were used to prepare the specimens—corrosive preparation, latex specimen preparation, and computer tomography imaging. This study revealed a configuration of the arterial circulation in the caracal encephalon resembling the shape of the number eight. The presence of the rostral communicating artery in this species is of particular significance, as it is associated with an increased ability to detect dehydration in the forebrain. This adaptation plays a crucial role in responding to challenges related to hydration. Comparative anatomical analysis with other felids highlighted differences in the shape and configuration of the encephalon’s arterial circulation. This study also discussed the obliteration of the extracranial segment of the internal carotid artery in adult caracals, a feature shared with other Felidae members. The results of this study provide valuable information regarding the anatomy of blood vessels in caracals, with potential implications for veterinary practice in zoos and wildlife conservation efforts. This research expands our knowledge of this species’ unique adaptations and physiological processes, contributing to the development of comparative anatomy in the Felidae family.

## 1. Introduction

The caracal belongs to the Felidae family of the order Carnivora [1,2]. It is found in forests, savannas, shrublands, grasslands, and deserts in Africa and Asia in the wild. Due to threats from hunting and the reduction of their natural habitat through land use for crops and livestock, this species has been placed on the Red List of the International Union for Conservation of Nature. The population trend is unknown, although it is very variable depending on the geographical area [3]. It is critically endangered in Pakistan [4] and close to extinction in many parts of northern Africa [5], while in southern and central Africa, the population appears stable [6]. Outside their natural habitat, caracals are maintained in captivity in zoos. Caracals reach a body mass of 5.8 to 22 kg. They are characterized by a relatively short tail for cats and long tufts of hair extending from their ears, reaching up to 8 cm in length [7]. Their fur is a uniform golden or reddish-brown shade, with a cream underbelly [8]. The average lifespan of a caracal is ten years, although individuals kept in captivity can live up to 19 years [7,9]. When hunting, caracals use vegetation as cover from detection. They prefer a predominantly nocturnal lifestyle, although there is a dependence on the ambient temperature [10].

The diet of caracals is mainly based on rodents and birds [11]. Additionally, they prey on small predators such as domestic cats [12,13]. Local ranchers often consider caracals pests because of their impact on livestock, especially poultry [14]. In South Africa, they are a target for hunting due to their attractiveness as trophies. There are reports of caracals being hunted for their meat. Notably, they are increasingly being treated as pets and kept for recreational purposes [15]. According to a study by Avenant and Nel [11], one caracal territory contains one male, four females, and between two and nine young.

Due to the uniqueness of the material, it was decided to perform a study of the encephalon base vessels, including the tributaries in this vascular area. A precise description of this area is of general biological and clinical significance. The information in this study may be helpful to veterinarians working in zoos or involved in rescuing wild animals in their natural environments. Knowledge of the anatomy of the encephalon vasculature is important for the correct interpretation of images obtained during diagnostic studies such as CT angiography. It may also be useful during surgical operations or minor procedures performed on the head [16,17]. The aim of this study was to analyze the arteries of the encephalon base in caracals.

## 2. Materials and Methods

### 2.1. Animals

This study was conducted on 14 preparations of animal heads and the encephalon arteries of both sexes of adult caracals (*Caracal caracal*) of the Felidae family. All animals utilized in this study were obtained from zoos as post-mortem specimens. No animals were intentionally euthanized or sacrificed for the purpose of this research. Approval from research ethics committees was not necessary to fulfill the objectives of this study, as the experimental procedures were solely performed on cadavers. We have provided more detailed information in this regard in the Institutional Review Board Statement.

### 2.2. Methods

In one of the methods implemented in three instances, the bilateral common carotid arteries were infused with red liquid latex (LBS 3060). Following the injection, the preparations underwent curing in a 5% formaldehyde solution for a duration of 14 days. Subsequently, the specimens were rinsed with running water for 48 h to remove any residual formaldehyde. The subsequent stage involved the manual dissection of soft tissues. The excess tissue was cut, which resulted in red arteries being obtained from the surrounding soft tissues. Extra safety measures were taken to ensure protection within the room designated for preparation. A ventilation system was in place, operating at 15 air changes per hour. The procedure commenced with the careful cutting of the skull bone using an oscillating saw, allowing access to the soft tissue. Subsequently, the blood vessels were meticulously prepared using surgical instruments. The preparation process was initiated with the removal of skin from the entire neck and head. Great care was taken during the preparation of muscle tissue to prevent any harm to the surrounding arteries. The next step involved delicately cleaning the prepared arteries of excess connective tissue.

In the next method, nine specimens underwent preparation that involved injecting red self-curing DURACRYL^®^ PLUS (SpofaDental, Jičín, Czech Republic) into the bilateral common carotid arteries. Following approximately 20 min for the specimens to cure, enzymatic maceration of the material with Persil powder (Henkel, Düsseldorf, Germany) at 42 °C continued for about one month. This process yielded corrosion casts of the blood vessels on the bone scaffold.

For the angio-CT examination, two cadavers were used. The specimens were frozen within 1 h of death. Examinations were performed immediately after thawing the specimens. Before initiating the scans, barium sulfate (*barium sulfuricum* 1.0 g/mL, Medana, Sieradz, Poland) was injected into the bilateral common carotid arteries. The heads were securely immobilized on a tomography table. Cone-beam computed tomography was used, with the following scanning parameters: 110 kVp, 0.08 mAs per shot, 20.48 mAs (Total mAs), and a reconstructed slice thickness of 0.29 mm.

The images were captured using a Nikon D3200 digital camera and saved in JPG format. Subsequently, the images underwent processing using “Preview Version 11.0 (1056.2.4), Copyright© 2002–2023 Apple Inc.” (Apple, Cupertono, CA, USA) digital image editing software. The names of the anatomical structures were standardized according to the *Nomina Anatomica Veterinaria* (2017) [18].

## 3. Results

The shape of the arterial circle of the encephalon resembles the shape of the number eight (Figure 1). The rostral part formed of the bilateral rostral cerebral arteries (*aa. Cerebri rostrales*) has gentle arcs, as does the caudal part formed of the bilateral caudal communicating arteries (*aa. Communicans caudales*).

The rostral cerebral artery arises from the intracranial segment of the internal carotid artery (*a. carotis interna*). Initially, it is placed on the base of the encephalon on the sides of the infundibulum of the hypophysis gland (*infundibulum glandulae pituitariae*), the mammillary body (*corpus mamillare*) and the tuber cinereum; it runs close to the optic chiasm (*chiasma opticum*) and goes to the longitudinal fissure of the brain (*fissure longitudinalis cerebri*). It also donates the internal ethmoid artery to the ethmoid plate of the ethmoid bone. There is no presence of a rostral communicating artery (*a. communicans rostralis*) in two cases, which causes the arterial circle of the encephalon to remain open from the nasal side. In the other cases, this vessel was observed (Figure 2). Rostrally, from the optic chiasm, the middle cerebral artery (*a. cerebri media*) branches off from the rostral cerebral artery. This is the strongest branch extending from the arterial circle of the encephalon. It surrounds the piriform lobe (*lobus piriformis*) from the rostral side and extends to the dorsolateral surface of the cerebral hemisphere (*facies dorsolateralis hemisheri*). Another branch of the rostral cerebral artery is the rostral choroid artery (*a. choroidea rostralis*). This is a thin vessel extending between the internal carotid artery and the middle cerebral artery.

The caudal part of the cerebral arterial circle is formed of the caudal communicating arteries, which, from their point of origin, follow a gentle arc, converging caudally and connecting to the odd basilar artery (Figure 3). From the caudal communicating artery, the caudal cerebral artery (*a. cerebri caudalis*) branches off, which is a single vessel. It usually divides into two branches, although in one case, the right side divides into two, and the left side divides into three branches. The division into branches falls close to the origin of this vessel. More caudally located are the rostral cerebellar arteries (*a. cerebelli rostrales*). These are strong vessels corresponding in diameter to the caudal arteries of the encephalon. The basilar artery (*a. basilaris*) is a vessel that maintains the same diameter along its entire course. Before an even portion of this vessel passes into the vertebral arteries (*aa. Vertebrales*), it gives half of its length to the caudal cerebellar arteries (*aa. Cerebelli caudales*). These vessels diverge asymmetrically, with the one on one side being slightly more rostral than that on the other side.

Among the tributaries contributing to the cerebral arterial circle, the rete mirabile of the maxillary artery should be considered first and foremost. Emerging from it is the intracranial segment of the internal carotid artery, which is the main source of blood flowing to the encephalon. The rete mirabile of the maxillary artery is a network of fine vessels branched off from the maxillary artery (Figure 4). It is located around this artery at the bottom of the orbit and heads towards the encephalon. The maxillary artery is the continuation of the external carotid artery (*a. carotis externa*), which is the follow-up of the common carotid artery (*a. carotis communis*). At the point where the common carotid artery evolves into the external carotid artery, the sinus carotis (*sinus caroticus*) is observed. The extracranial segment of the internal carotid artery that branches off from the sinus carotis is not observed. In addition, the vertebral arteries (*a. vertebrales*), which connect to the basilar artery, are the second source of blood flow to the cerebral arterial circle (Figure 4).

## 4. Discussion

An analogous shape of the cerebral arterial circle, as in the described species, has been found in other representatives of the Felinae subfamily, e.g., cat (*Felis catus*), serval (*Leptailurus serval*), Eurasian lynx (*Lynx lynx*), jungle cat (*Felis chaus*), puma (*Puma concolor*), and leopard cat (*Prionailurus bengalensis*) [19]. In contrast, in the described representatives of the Pantherinae subfamily, e.g., lion (*Panthera leo*), leopard (*Panthera pardus*), jaguar (*Panthera onca*), and tiger (*Panthera tigris*), the rostral part of the circle is more elongated and less rounded, while the caudal part is proportionally more diminutive and more elongated, with less protrusion to the sides [19]. The authors cited above compared the rostral segment of the cerebral arterial circle in Felinae to the letter C, and in Pantherinae, they compared it to the letter J. In contrast to some animal groups, the cerebral arterial circle in giraffes takes on a triangular shape [20] or a heart shape in other ruminants [21,22,23,24]. Those of camels are similar to the number “8” [25,26,27,28]. In Caniformia (the second suborder in the Carnivora beyond the Feliformia), e.g., the dog and fox, is elliptical in shape [29,30]. In most of the Felidae described, the caudal connecting arteries are connected by the *ramus anastomoticus circuli arteriosi* [19]. Such a connection was not found in the caracal.

In the analysis of the specimens, the notable presence of the rostral communicating artery was observed. In a study conducted by Fenrich in 2021 [31], it was established that this artery plays a pivotal role in the detection of dehydration in the forebrain. According to Fenrich’s findings, animals equipped with this specific artery have a significant edge in identifying and responding to dehydration. Dehydration, marked by the loss of water, triggers a decrease in extracellular fluid, resulting in reduced tissue perfusion and a subsequent shift towards anaerobic metabolism in the peripheral tissues. Considering the encephalon’s limited capacity for anaerobic metabolism, it assumes a privileged status during dehydration. To mitigate and postpone anaerobic metabolism within the encephalon, a sufficient blood supply is of utmost importance. Placing dehydration sensors near the neurons responsible for relevant effector homeostatic mechanisms proves exceptionally advantageous for the efficiency of this physiological process [31].

A single caudal cerebral artery has been found in jaguars, leopards, lynxes, tigers, and domestic cats. This vessel, in a double form, has been described in lions and servals [19]. In the caracal, the rostral cerebellar artery was observed in a single form, while in other felids, it was a double vessel [19]. The caudal cerebellar artery originates as a branch from the basilar artery. The initial part of this vessel departs at a 90-degree angle. Such a model of branching off suggests that blood to this vessel could flow into the caudal cerebellar artery, both from the caudal communicating artery side and from the vertebral artery side [32]. As in the caracal, a strong basilar artery that does not decrease in diameter is found in other felids, canids, and camels [19,27,33,34,35]. An additional feature that may support this claim is that the basilar artery has the same diameter over its entire length, i.e., its diameter does not decrease in the caudal direction. An experiment by Baldwin and Bell [36,37,38] on sheep and cattle showed that the basilar artery, which decreases in diameter in the caudal direction, is not involved in supplying blood to the encephalon in these species.

In the caracal, the extracranial segment of the internal carotid artery is obliterated, as in other Felidae [19,33,39,40,41,42,43]. In young cats, the internal carotid artery is preserved along its entire length. That is, there is an unobstructed extracranial segment of this vessel. Observations of fetuses show it as a strong vessel supplying blood to the encephalon. In specimens of about 4–8 weeks of age, there is a gradual atrophy of the extracranial segment of this vessel, which is absent in adult animals [43]. This study was only conducted on adult specimens, so no case had an extracranial segment of the internal carotid artery. In Caniformia, the internal carotid artery is well developed and does not undergo obliteration. Studies conducted on the raccoon dog (*Nyctereutes procyonoides*), red fox (*Vulpes vulpes*), grey wolf (*Canis lupus*), American mink (*Mustela vison*), European badger (*Meles meles*), Eurasian otter (*Lutra lutra*), common raccoon (*Procyon lotor*), and grey seal (*Halichoerus grypus*) present the occurrence and course of this vessel in adults [35,44,45,46,47,48].

Similar to the caracal, the phenomenon of the obliteration of the extracranial segment of the internal carotid artery has been described in ruminants [22,23,49,50,51,52,53]. On the base of the encephalon in this group of animals, the rostral epidural rete mirabile (*rete mirabile epidurale rostrale*) is present. It is similar to the rete mirabile of the maxillary artery found in the caracal and other felids, but located on the bottom of the cranial cavity. In fetuses and juveniles of cattle and Eurasian elk [21,24], a fully preserved internal carotid artery has been found as one of the sources of blood to the rostral epidural rete mirabile, and consequently, to the encephalon. As these animals grew, the extracranial segment of this vessel was found to be obliterated. In adults, the source of blood to the rostral epidural rete mirabile was the branches diverging from the maxillary artery. Thus, in terms of supplying blood to the encephalon, both retia have the same function, although their locations are different.

## 5. Conclusions

The results of this study revealed that the caracal’s arterial circle of the encephalon resembles the shape of the number eight. The presence of the rostral communicating artery is probably associated with the caracal’s increased ability to detect dehydration in the forebrain. Comparative anatomical analysis with other members of the Felidae family highlighted differences in the shape and configuration of the encephalon’s arterial circulation. The obliteration of the extracranial segment of the internal carotid artery in adult caracals is a common feature among other members of the Felidae family.

## Figures and Tables

**Figure 1 animals-13-03780-f001:**
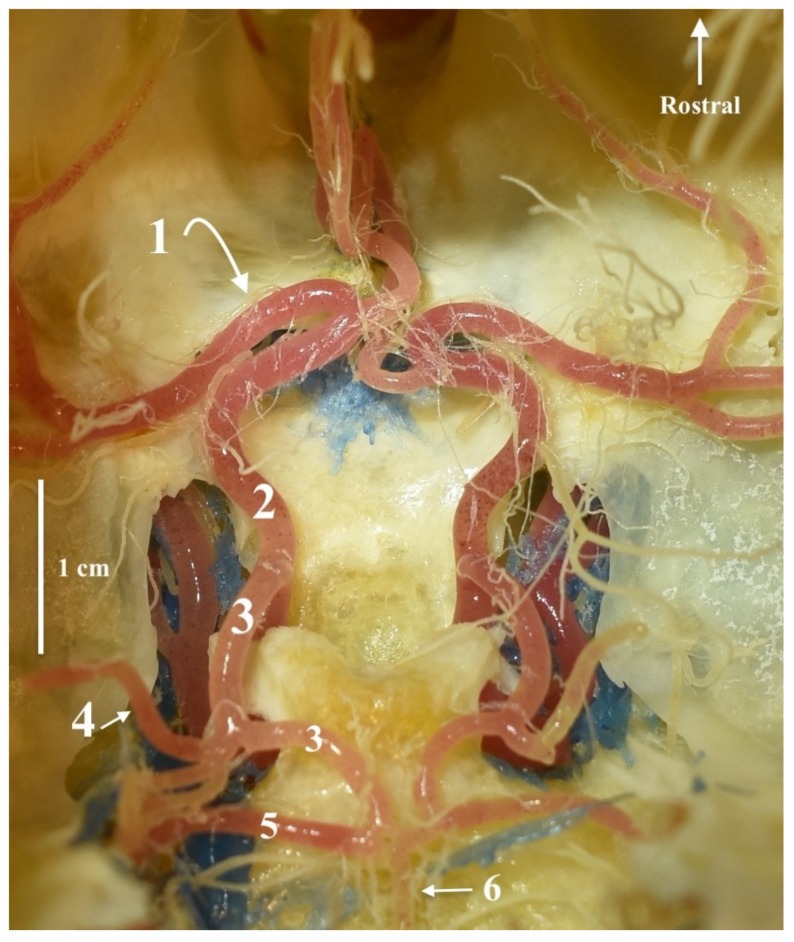
The cerebral arterial circle. Dorsal view. Corrosion cast. 1—the middle cerebral artery; 2—the rostral cerebral artery; 3—the caudal communicating artery; 4—the caudal cerebral artery; 5—the rostral cerebellar artery; 6—the basilar artery.

**Figure 2 animals-13-03780-f002:**
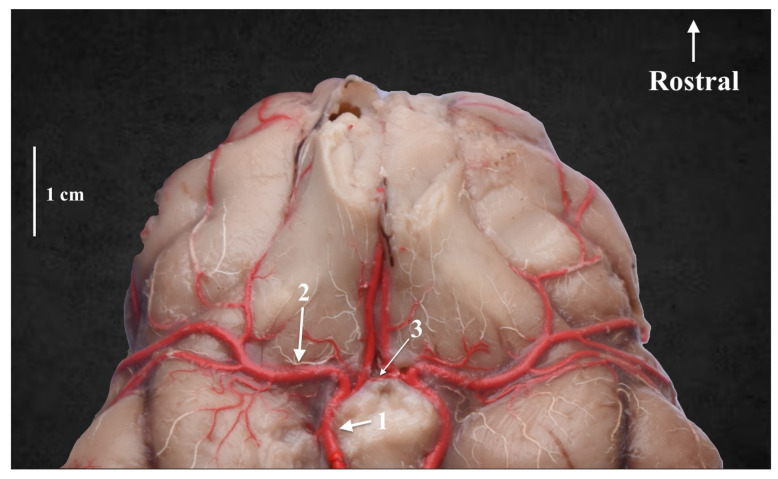
The rostral part of the cerebral arterial circle. Ventral view. Latex preparation. 1—the rostral cerebral artery; 2—the middle cerebral artery; 3—the rostral communicating artery.

**Figure 3 animals-13-03780-f003:**
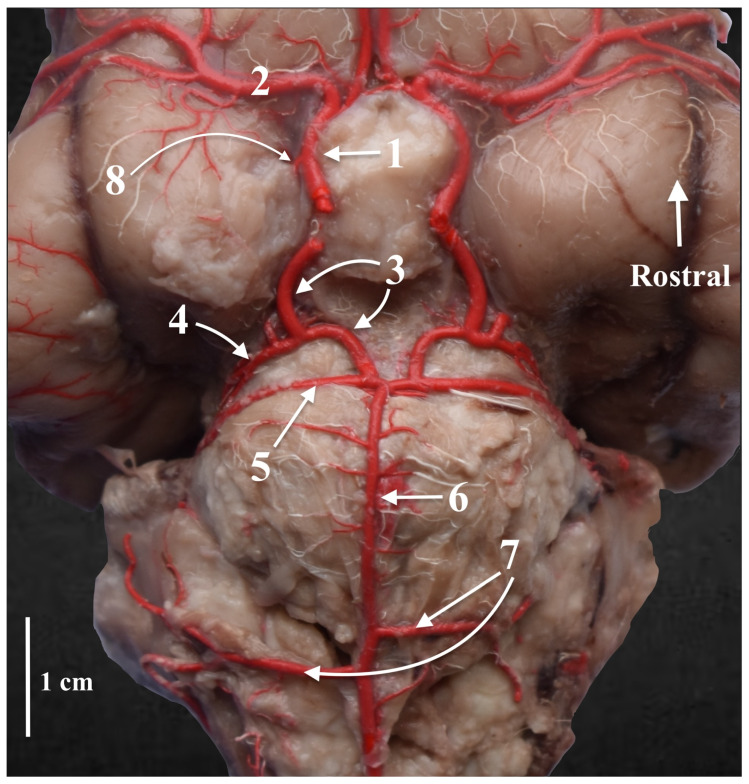
The cerebral arterial circle. Ventral view. Latex preparation. 1—the rostral cerebral artery; 2—the middle cerebral artery; 3—the caudal communicating artery; 4—the caudal cerebral artery; 5—the rostral cerebellar artery; 6—the basilar artery; 7—the caudal cerebellar artery; 8—the rostral choroid artery.

**Figure 4 animals-13-03780-f004:**
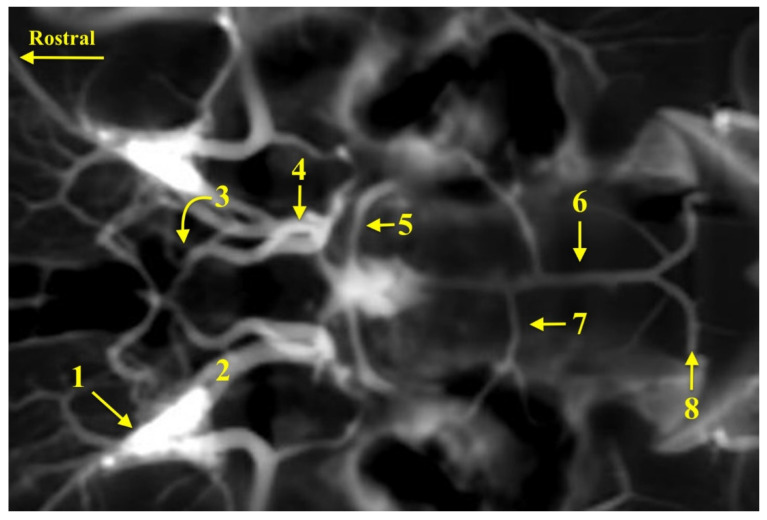
The angio-CT scan of the head of a caracal. 1—the rete mirabile of the maxillary artery; 2—the intracranial segment of the internal carotid artery; 3—the rostral cerebral artery; 4—the caudal communicating artery; 5—the rostral cerebellar artery; 6—the basilar artery; 7—the caudal cerebellar artery; 8—the vertebral artery.

## Data Availability

The data presented in this study are available on request from the corresponding author.

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
