# Peer review of "The Arteries of the Encephalon Base in Caracal (Caracal caracal; Felidae; Carnivora)"

_animals, 2023, doi:10.3390/ani13243780_

Round 1

Reviewer 1 Report

Comments and Suggestions for Authors

The manuscript describes the ventral arterial supply to the encephalon, such as the cerebral arterial circle and its tributaries. The figures are very well presented and the anatomical description well performed. The article can be accepted after the below corrections and suggestions:

Include the family and the order inside the parenthesis in the title.

Change the word brain to encephalon along the manuscript

Line 34: Delete the Felidae and Carnivora that are inside the parentheses and include a more recent phylogenetic reference in carnivorans.

Line 102: change the term pituitary to hipophysis based on the NAV

114: inner jugular?

154 and 156: include the scientific names of those Felids.

165: such as?

168 and 216: forebrain or telencephalon?

181: in other felids...

186: felids, canids, ...

199-201 : include the scientific names

218-209: you do not include that you discussed because it is a conclusion. The obliteration of the extracranial segment of the internal carotid artery in adult caracals is a feature shared with other members of the family Felidae. 

References

Correct the author of reference 8: International committee on veterinary gross anatomical nomenclature.

Reviewer 2 Report

Comments and Suggestions for Authors

Very interesting paper. The arterial circle is very similar to that found in the common cat. 

However, I will point out a few comments to be answered:

-           In line 103 you should better use the term optic chiasm instead of optic junction.

-           In line 105 you have not found a rostral communicating artery in two cases. However you may have missed it. In dog, several fine anastomosis exists between the lefty and right rostral cerebral arteries. If you have found it in other cases, they should be present.

-           In lines 112 and 113 you say that the rostral choroid artery is a branch of the rostral cerebral artery. You have not shown that in an image of a dissection and it should have been good to do it because in dog, the rostral choroid artery branches off from the beginning of the middle cerebral artery or from the beginning of the caudal communicating arteries.

-           In line 113 there is a sentence that I do not understand: “It is a thin vessel extending halfway between 113 the inner jugular and the middle of the brain”.

-           In lines 135 and 136 you say: “Emerging from it is the in intracranial segment of the internal carotid artery, which is the main source of blood flowing to the brain”. Another contribution to the intracranial segment of the ICA, in the cat, is the ascending pharyngeal artery through the carotid foramen.

-           In lines 141 to 144 your statements should have images to prove them because in cat, the carotid body and carotid sinus are located at the beginning of the occipital artery. Also in cat, the external segment of the ICA is a branch of the occipito-ascending pharyngeal artery. Moreover, in cats and artiodactyls (pig and ruminants), the blood flow of the basilar artery follows a rostro-caudal direction. So, it does not contribute to the vascularization of the encephalon.

-           In lines 188 to 190 the above issue comes again. May be in the Caracal, the blood flow runs in caudo-rostrally but this should have been commented and proved may be in a following publication.

-           In lines 201 to 203 the obliteration of the external segment of the ICA takes also place in pigs. It seems that the animals with a “rete mirabile” follow the same pattern. So the same should apply for the basilar artery.

In Figure 2, the number 3 should be equivalent to the dog rostral intercarotid artery for vascularization of the neurohypophysis.

Reviewer 3 Report

Comments and Suggestions for Authors

The paper deals with the vascularization of the caracal brain. The work has both cognitive and clinical value. The conducted research is fully justified.
In the introduction, please describe in more detail the ecology of these animals, i.e. what animals mainly fall prey to them, whether they hunt at night or during the day and how.

In Materials and methods, please specify how much time passed between death and CT imaging (especially since injections were performed).

The photos of the injection preparations are very nice and are a valuable addition to the text.

The discussion is maire extensive and well supported.
The literature is sufficient.
